# Quantitative Determination of Biogenic Element Contents and Phytochemicals of Broccoli (*Brassica oleracea* var. *italica*) Cooked Using Different Techniques

**DOI:** 10.3390/plants13101283

**Published:** 2024-05-07

**Authors:** Fahad AlJuhaimi, Isam A. Mohamed Ahmed, Mehmet Musa Özcan, Nurhan Uslu, Zainab Albakry

**Affiliations:** 1Department of Food Science and Nutrition, College of Food and Agricultural Sciences, King Saud University, PO Box 2460, Riyadh 11451, Saudi Arabia; faljuhaimi@ksu.edu.sa (F.A.); iali@ksu.edu.sa (I.A.M.A.); 2Department of Food Engineering, Faculty of Agriculture, Selcuk University, Konya 42031, Turkey; nurhanuslu.gmuh@gmail.com; 3College of Ocean Food and Biological Engineering, Jimei University, Xiamen 361021, China; 20236100007@jmu.edu.cn

**Keywords:** broccoli, cooking methods, bioactive compounds, antioxidant capacity, polyphenols, elements, HPLC

## Abstract

In this study, the effect of different cooking techniques on broccoli moisture, total phenolic, total flavonoid, and radical scavenging capacity results, polyphenol contents, and their quantitative values was investigated. The total phenolic quantities of fresh and cooked broccoli samples were assessed to be between 36.32 (conventional boiling) and 423.39 mg GAE/100 g (microwave heating). The radical scavenging activities of the broccoli samples were reported between 2.55 (conventional boiling) and 4.99 mmol/kg (microwave heating). In addition, catechin and rutin quantities of the fresh and cooked broccoli samples were measured to be between 2.24 (conventional boiling) and 54.48 mg/100 g (microwave heating), and between 0.55 (conventional boiling) and 16.33 mg/100 g (microwave heating), respectively. The most abundant elements in fresh and cooked broccoli samples were K, Ca, P, S, and Mg. The results showed some changes depending on cooking techniques compared to the control. The bioactive properties of broccoli samples cooked by means of conventional boiling, boiling in vacuum bag, and high-pressure boiling were established to be lower compared to the fresh sample. Catechin, 3,4-dihydroxybenzoic acid, rutin, and gallic acid were the key phenolic compounds of fresh and cooked broccoli samples. The phenolic components of broccoli were significantly affected by the applied cooking techniques. The highest protein in broccoli samples was determined in the broccoli sample cooked by boiling in a vacuum bag. There were statistically significant changes among the mineral results of broccoli cooked with different cooking methods.

## 1. Introduction

Broccoli (*Brassica oleracea* var. *italica*), which is native to the Eastern Mediterranean basin and Italy, and a member of the Brassicaceae family, is a horticultural and a favorite winter vegetable with high nutritional value, bioactive compounds, and antioxidant properties [1,2,3,4,5]. Phenolic compounds, phenolic acids such as kaempferol and ascorbic acid in broccoli have antioxidant capacity [6,7], Broccoli contains significant amounts of health-beneficial compounds, which has increased its consumption by people [8]. The consumption of broccoli has a positive effect on human health thanks to some minerals, phenolics, xanthophylls, sulforaphane, phenolics, and unique bioactive compounds [6]. Broccoli contains significant amounts of health-beneficial compounds, which has increased its consumption by people. Minerals are the building blocks of bones, teeth, blood, and muscle cells, and increase the usefulness of vitamins [9,10,11]. Phenolic compounds play a crucial role in exerting a wide variety of biochemical and pharmacological effects [12,13].

The phenolic components of foods, and therefore their potential antioxidant activities, are affected by domestic processing procedures such as cooking [14,15] and thermal treatments [16,17]. If oxidative enzymes are not inactivated during cooking, they can increase the chemical or enzymatic degradation of phenolic compounds or cause chemical changes that may affect quality properties [18,19]. Steaming has been reported to be more beneficial for certain health-promoting compounds in fruits and vegetables [20]. In addition to causing changes in the chemical composition of green leafy vegetables, cooking processes can also cause changes in the concentrations of bioactive compounds [21,22,23,24]. Among the main factors that can change the level of phytochemicals in vegetables before consumption, the most important ones are heat-involving preparation/processing, thermal degradation, oxidation, leaching, and matrix degradation [25]. Broccoli cooked with various cooking methods can also be used in salads or side dishes. Although the most common cooking methods are usually boiling and steaming, the use of sous vide has recently become widespread in order to preserve the content and increase the bioavailability of bioactive compounds in vegetables [20,26,27]. Various heat treatments are widely applied to green leafy vegetables, which are commonly consumed raw or cooked/processed, to neutralize microorganisms and enzymes and increase flavor, thus increasing product safety and quality [26,28]. Vegetables and fruits are foodstuffs that have an important place in nutrition. They contribute to human health not only with the vitamins and minerals they contain, but also with their phenolic components. Combined phytochemicals in plant foods act through various mechanisms such as antioxidant activity, cell regeneration, and tumor suppression [29,30]. Vegetables are consumed raw as well as processed and consumed as various products. Since it is difficult for vegetables to be stored for a long time without spoiling in fresh form, it is possible to store them for a long time by applying processes such as boiling and drying [31]. Boiling is applied to prevent enzymatic change in canned food production until heat treatment, to prevent the negative effects of enzymes in the drying process until the end of drying, and to prevent the effect of enzymes until consumption in freezing preservation. Enzymes are rendered inactive by these processes. Thus, the raw material is prevented from undergoing enzymatic changes until it is sterilized. In addition, the microorganism load is also reduced by boiling [32]. Drying is one of the first preservation methods used by humanity to preserve food. Today, in parallel with the increasing consumption trend of ready-made foods, the importance of dried vegetables, which is one of the basic ingredients of such foods, is increasing all over the world. Although vegetables are generally consumed as fresh in Turkey, vegetables dried by various methods are also demanded by final consumers and food industry companies. The dried vegetable sector has become one of the important sub-sectors of the food industry with the modern drying methods it uses as well as the traditional sun drying method [30,33]. Boiling and steaming are the most widely used traditional cooking methods. Traditional cooking methods can lead to a loss of nutrients and flavor elements [34]. New cooking methods are being studied to minimize these losses. The objective of this investigation was to monitor the effect of different cooking techniques on broccoli moisture, total phenolic, total flavonoid, and radical scavenging capacity results, polyphenol contents, and their quantitative values.

## 2. Results and Discussion

### 2.1. Total Phenolic and Total Flavonoid Amounts, and Antioxidant Activity Values of Broccoli Cooked with Different Cooking Techniques

The moisture quantities and bioactive properties of broccoli florets cooked with different cooking techniques are displayed in Table 1. The results showed some changes depending on cooking techniques compared to the control. The moisture quantities of fresh and cooked broccoli samples were found to be between 50.84 (microwave) and 91.82% (conventional boiling). In general, the moisture quantities of broccoli cooked by means of conventional boiling (open pot), boiling in a vacuum bag, and high-pressure boiling were established to be higher when compared to fresh, while the moisture amounts of broccoli samples cooked with conventional heating and microwave heating application are found to be lower than those of fresh ones.

While the total phenolic amounts of cooked broccoli samples were found to be between 36.32 (conventional boiling) and 423.39 mg GAE/100 g (microwave heating), the total phenolic quantity of fresh broccoli samples was 116.69 mg GAE/100 g. Also, the total flavonoid quantity of fresh broccoli samples was recorded as 157.30 mg/100 g, while the total flavonoid quantities of cooked broccoli samples ranged between 60.16 (conventional boiling) and 409.68 mg/100 g (microwave heating). In addition, the antioxidant activities of fresh and cooked broccoli samples were assessed to be between 2.55 (conventional boiling) and 4.99 mmol/kg (microwave heating). While the total phenol and total flavonoid quantities and the antioxidant activities of broccoli samples cooked by means of conventional boiling, boiling in a vacuum bag, and high-pressure boiling were found to be lower compared to the fresh sample, the total phenol, total flavonoid, and antioxidant activities of broccoli samples cooked in conventional heating and microwave heating were established to be higher than the results of both fresh broccoli and broccoli cooked using the other three cooking techniques. The fact that the total phenol and total flavonoid quantities, and the radical scavenging capacity values of broccoli samples cooked using conventional and microwave heating were higher than the others is probably caused by Maillard reaction products that may occur as a result of dry heating. The amount of total phenols and flavonoids in broccoli samples cooked by means of boiling in a vacuum bag, high-pressure boiling, and conventional boiling may have decreased due to the deterioration of their structure. There were statistically significant changes among the bioactive characteristics of broccoli cooked with different cooking methods (*p* < 0.05). Many physical and chemical changes and thermal deteriorations may occur in the structure of most vegetables cooked using boiling, microwave oven, steaming, or baking methods [35]. In addition, Maillard reaction products, formed as a result of heat treatment, can produce stronger antioxidant products [36,37]. The moisture contents of fresh broccoli, water-boiled broccoli, steamed broccoli, and microwaved broccoli were 89.86%, 93.26%, 91.87%, and 93.27%, respectively [38]. The protein contents (dw) of fresh broccoli, water-boiled broccoli, steamed broccoli, and microwaved broccoli were 3.34%, 2.27%, 2.68%, and 2.26%, respectively [38]. In addition to these, the total phenolic content of broccoli ranged from 412 to 987 mgGAE/100 g in fresh samples [34,39]. The total phenolic contents of raw broccoli and broccoli cooked using MW, boiling, and steaming were 169.6 mg GAE/100 g, 164.3–185.8 mgGAE/100 g, 164.2–171.3 mgGAE/100 g, and 67.9–139.3 mgGAE/100 g, respectively [40]. The total phenolic content and radical scavenging activity of raw broccoli were found to be 2282.97 mgGAE/kg and 0.189 mmolTE/g, respectively [5]. The total phenolic result and radical scavenging activities of raw broccoli cooked using the sous vide method at different times (5, 10, and 15 min.) ranged between 1845.88 and 2098.96 mg GAE/kg, and between 0.119 and 0.158 mmolTE/g, respectively [5]. The total phenolic and antioxidant activity results of raw broccoli cooked using the steaming method at different times (5, 10, and 15 min.) ranged between 1981 and 2188.3 mg GAE/kg, and between 0.144 and 0.174 mmolTE/g, respectively [5]. The total phenolic content and antioxidant activities of raw broccoli cooked using the boiling method at different times (5, 10, and 15 min.) ranged between 926.56 and 1692.47 mgGAE/kg, and between 0.069 and 0.133 mmolTE/g, respectively [5]. The total phenolic content of vegetables decreased using both conventional and sous vide cooking methods [41]. The antioxidant activity values of raw broccoli and broccoli cooked using MW, steaming, and boiling were 637, 563–692, 599–732, and 249–617 μmol TE/100 g, respectively [40]. Turkmen et al. [42] pointed out that the cooking method with the highest total phenol content of broccoli was microwave cooking. In a previous study, there were no differences in antioxidant activity between fresh and cooked broccoli [43]. It has been stated that there are differences even in the antioxidant capacity results of different parts of broccoli [44,45]. Findings pointed out some fluctuations compared to the results of several studies. These changes in results are probably due to the genetic variation of the sample, different climatic factors, plant parts, boiling times and types, agronomical conditions, harvest time, variety, and analytical conditions such as solvent used and extraction types.

### 2.2. The Phenolic Compounds of Fresh and Cooked Broccoli Samples

The phenolic profiles and their quantitative results of fresh and cooked broccoli samples are displayed in Table 2. Catechin, 3,4-dihydroxybenzoic acid, rutin, and gallic acid were the key phenolic compounds of fresh and cooked broccoli samples (Figure 1). It was observed that the phenolic components of broccoli were significantly affected by the applied cooking techniques. It was determined that the amount of phenolic components of cooked broccoli significantly decreased when compared to the fresh sample. However, the amounts of some phenolic profiles differed depending on the cooking technique. The phenolic components of broccoli samples cooked with conventional heating and microwave heating were higher when compared to the results of fresh broccoli and broccoli cooked using other cooking techniques. While the gallic acid quantities of broccoli florets varied between 0.52 (conventional boiling) and 4.29 mg/100 g (microwave heating), the 3,4-dihydroxybenzoic acid quantities of broccoli samples were assessed to be between 0.41 (conventional boiling) and 15.17 mg/100 g (microwave heating). In addition, the catechin and rutin quantities of fresh and cooked broccoli samples were assessed to be between 2.24 (conventional boiling) and 54.48 mg/100 g (microwave heating), and between 0.55 (conventional boiling) and 16.33 mg/100 g (microwave heating), respectively. The highest caffeic (7.07), syringic (6.21), *p*-coumaric acid (2.03), ferulic acid (4.72), resveratrol (1.11), quercetin (4.68), cinnamic acid (0.78), and kaempferol (5.12 mg/100 g) were found in broccoli cooked in the microwave. A significant part of the phenolic compounds in broccoli were adversely affected by conventional boiling, boiling in a vacuum bag, and high-pressure boiling methods. The catechin contents of broccoli cooked by means of boiling in a vacuum bag, high-pressure boiling, and conventional boiling were higher than broccoli cooked using conventional boiling. There were statistically significant changes among the phenolic compounds of broccoli cooked with different cooking methods (*p* < 0.05). Gunathilake et al. [46] pointed out that there is a decrease in total polyphenolic compounds during the cooking of some vegetables, and this decrease is probably due to the diffusion of polyphenolic compounds into boiling water. The effect of cooking methods on the release of phenolic constituents showed that heat treatments cause a partial hydrolysis of conjugated polyphenols. The release of polyphenols into free phenolic compounds triggered by heat treatment has been reported [23,35]. Caffeic acid, chlorogenic acid, and neochlorogenic acid were the major phenolic acids found in broccoli [45,47]. Fresh broccoli and broccoli cooked in the microwave contained 1.845 and 0.173 µg/g quercetin, 1.230 and 0.474 isorhamnetin, 4.976 and 7.252 *trans*-ferulic acid, 0.262 and 1.258 *p*-coumaric acid, and 52.158 and 46.489 µg/g chlorogenic acid, respectively [48]. Lopez-Hernandez et al. [40] reported that 0.27 mg/kg gallic acid, 16.43 chlorogenic acid, 0.97 caffeic acid, 10.89 isoquercitrin, 0.16 myricetin, 0.30 luteolin, and 0.12 quercetin were identified in raw broccoli. Also, 0.28–0.32 and 0.29–0.35 mg/kg gallic acid, 11.97–18.76 and 15.25–25.84 chlorogenic acid, 0.72–1.06 and 0.93–1.27 caffeic acid, 10.49–11.16 and 9.87–12.86 isoquercitrin, 0.14–0.18 and 0.14–0.18 myricetin, 0.31–0.37 and 0.39–15.01 luteolin, and 0.10–0.13 and 0.11–0.15 mg/kg quercetin were detected in broccoli cooked using MW and steaming, respectively [40]. In another study, adequate amounts of chlorogenic, neochlorogenic, and ferulic acids were detected in three broccoli samples (stem, leaf, and flower), while caffeic and *p*-coumaric acids were measured in broccoli leaf extracts. In addition, gallic acid and vanillic acid were detected in broccoli stems and flower extracts, while sinapic acid was detected only in broccoli leaves and flower extracts [49]. Our findings illustrated some fluctuations compared to the results of previous studies. These changes are likely due to broccoli variety, agricultural and climatic factors, harvest time, used parts, solvent types, extraction methods, and some other factors such as analytical conditions, storage, and cooking times and types.

### 2.3. Mineral and Protein Contents of Broccoli Cooked with Different Cooking Techniques

The mineral and protein results of fresh and cooked broccoli samples are depicted in Table 3. The most abundant elements in fresh and cooked broccoli samples were K, Ca, P, S, and Mg. The element with the highest amount among the microelements was Fe, followed by Zn, Mn, B, and Cu in decreasing order. While the P amounts of fresh and cooked broccoli samples ranged between 3114.94 (fresh) and 4383.36 mg/kg (conventional boiling), the K quantities of broccoli samples were assessed to be between 19,638.27 (conventional boiling) and 32,729.06 mg/kg (boiling in vacuum bag). Also, the Ca and Mg amounts of fresh and cooked broccoli samples were found to be between 3555.67 (fresh) and 6052.mg/kg (conventional boiling), and between 1197.02 (fresh) and 5039.27 mg/kg (conventional heating), respectively. In addition, the S quantities of fresh and cooked broccoli samples varied between 3167.03 (conventional boiling) and 5112.97 mg/kg (microwave heating). Looking at the microelements, the Fe and Zn quantities of fresh and cooked broccoli samples were assessed to be between 41.46 (boiling in vacuum bag) and 55.99 mg/kg (conventional heating), and between 11.88 (fresh) and 17.53 mg/kg (conventional heating), respectively. Also, while the Cu results of fresh and cooked broccoli samples ranged between 2.56 (conventional heating) and 3.55 mg/kg (boiling in vacuum bag), the Mn amounts of broccoli samples were measured to be between 9.61 (fresh) and 15.22 mg/kg (high-pressure boiling). In general, the mineral content of broccoli samples cooked by different methods increased when compared to the control (fresh). Partial reductions were observed in some of the applied cooking techniques. The protein contents of fresh and cooked broccoli samples ranged between 13.35% (fresh) and 17.27% (boiling in a vacuum bag). Therefore, the highest protein in broccoli samples was determined to be in the broccoli sample cooked by means of boiling in a vacuum bag, followed by the sample cooked using conventional boiling, high-pressure boiling, microwave heating, conventional heating, and the fresh sample in descending order. There were statistically significant changes among the mineral results of broccoli cooked with different cooking methods (*p* < 0.05). Broccoli is a good source of elements such as Ca, Mg, Na, K, Ca, Cl, P, and S, and trace elements such as Fe, Zn, Mn and Cu, which are essential for human nutrition [7,50]. Fresh broccoli, water-boiled broccoli, steamed broccoli, and microwaved broccoli contained 8.67, 8.01, 8.11, and 8.61 mg/100 g Zn; 2.66, 1.68, 2.33, and 1.78 mg/100 g Fe; 112.52, 28.52, 94.21, and 63.79 mg/100 g Ca; 562.22, 275.37, 447.72, and 205.20 mg/100 g Mg; 3992.2943.76, 3796.3, and 2460.02 mg/100 g K; 576.52, 235.45, 379.58, and 256.61 mg/100 g Na; respectively [38]. Fresh broccoli contained 562.22 Mg, 3992.4 K, 576.52 Na, 8.67 Zn, 2.66 Fe, and 112.52 Ca mg/100 g (dw) [38]. Our results showed some differences with results described by Farnham et al. [50], Mukherjee et al. [7], and Mansour et al. [38], who stated that broccoli is a good alternative source of Ca, K, and Na. It was thought that the bioactive properties, phenolic compounds, and mineral and protein quantities of fresh and cooked broccoli samples probably vary depending on growing conditions, the harvest time of broccoli, soil plant nutrient elements, cooking techniques, cooking time, and analytical conditions.

## 3. Material and Methods

### 3.1. Material

Broccoli samples were purchased from a local market in Konya province in Turkey. The samples were brought in cool conditions to the laboratory, washed, and divided into florets. The broccoli florets had stems and were approximately 3–4 cm wide and 6–7 cm long.

### 3.2. Methods

#### 3.2.1. Boiling and Heating Processes

Broccoli samples were cooked using conventional boiling, boiling in a vacuum bag, a pressure cooker, conventional heating, and microwave heating for 13, 13, 7, 10, and 10 min. In the pressure cooker, 7 min was taken into account as the cooking time from the steam exit. In the oven, after the temperature was adjusted to 200 °C, a cooking time of 10 min was applied. The sous vide process was carried out using 100 g broccoli in plastic packaging at 100 °C/10 min. Conventional boiling was carried out in an open pot. Also, broccoli samples were heated in a microwave at 720 W for 10 min.

#### 3.2.2. Determination of Moisture Results of Broccoli Samples

The moisture results of broccoli samples were recorded using the KERN & SOHN GmbH infrared moisture analyser.

#### 3.2.3. Determination of Protein Quantities of Broccoli Samples

The protein contents of the broccoli samples were established according to the AOAC [51] method.

#### 3.2.4. Extraction Procedure

Broccoli samples were extracted according to the study recognized by Girgin and El [52]. After 3 g powdered broccoli samples was added to 20 mL of solvent (methanol–water, 80:20, *v*/*v*), the solution was stored in an ultrasonic bath for 30 min. Then, it was centrifuged for 10 min. The supernatant was removed and these steps were carried out twice with 20 mL of solvent. The combined extracts were used for analyses.

#### 3.2.5. Total Phenolic Results of Broccoli Samples

The Folin–Ciocalteu (FC) reagent was used to determine the total phenolic contents of broccoli extract according to Yoo et al. [53]. FC (1 mL) and Na_2_CO_3_ (10 mL) were added to the extract and mixed using a vortex mixer. Deionized water was added until the final volume was 25 mL, and the sample was kept in darkness for 1 h. After pre-processing, the absorbance value of the sample was recorded at 750 nm. The findings are stated as mg gallic acid equivalent/100 g.

#### 3.2.6. Total Flavonoid Content of Broccoli Samples

The total flavonoid results of broccoli samples were obtained according to the work recognized by Hogan et al. [54]. After pre-processing, the absorbance result of the mixture was measured at 510 nm. The findings obtained are given as mg quercetin (QE)/100 g.

#### 3.2.7. Antioxidant Activity Results of Broccoli Samples

1.1-diphenyl-2-picrylhydrazyl (DPPH) was applied for the antioxidant activity results of broccoli extracts [55]. The extract was added to 2 mL of a methanolic solution of DPPH, and was then vortexed and kept in darkness for 30 min; the absorbance of extracts was obtained at 517 nm. The results obtained are given as mmol trolox (TE)/kg.

#### 3.2.8. Phenolic Compounds

A High Performance Liquid Chromatography unit mounted on a PDA detector and an Inertsil ODS-3 (5 µm; 4.6 × 250 mm) column were used for the chromatographic separation of the phenolic compounds of broccoli samples. The injection volume was 20 µL. The peaks were recognized at 280 using a PDA detector. The mobile phase was a mixture of 0.05% acetic acid in water (A) and acetonitrile (B) with a flow rate of 1 mL/min at 30 °C.

#### 3.2.9. Determination of Minerals of Broccoli Samples

After 0.5 g broccoli powder, dried at 70 °C, was incinerated by using 5 mL of 65% HNO_3_ and 2 mL of 35% H_2_O_2_ in a closed microwave, its volume was completed with 20 mL distilled water. Minerals were measured using ICP-AES [56].

### 3.3. Statistical Analyses

The JMP version 9.0 statistical analysis method was used for the analysis of variance. The mean of the triplicate analysis data was subjected to analysis of variance. The significant differences among the values of control and cooking types were determined using Duncan’s Multiple Range Test (*p* < 0.05).

## 4. Conclusions

Cooking methods affect the bioactive substances, phenolic profiles, mineral result, and antioxidant activity observed in broccoli. The total phenol, total flavonoid, and radical scavenging capacity results of broccoli samples cooked using conventional boiling, boiling in a vacuum bag, and boiling in a steam cooker were found to be lower compared to the fresh sample. The conventional and microwave heating methods can be recommended as heat treatments that better preserve the original content of beneficial substances in broccoli. More studies are needed to preserve the content of health-promoting nutrients in consumed broccoli, gain new insights, and optimize the way broccoli is cooked. Cooking techniques and times had significant effects on the bioactive components, antioxidant activities, polyphenol contents, mineral results, and protein values of broccoli. The bioactive properties of broccoli samples cooked using conventional heating and microwave heating were established to be higher than the results of both fresh broccoli and broccoli cooked using the other three cooking techniques. The phenolic constituents of broccoli samples cooked with conventional heating and microwave heating were higher when compared to the results of fresh broccoli and broccoli cooked using the other cooking techniques. Therefore, the highest protein in broccoli samples was determined to be in the broccoli sample cooked by means of boiling in a vacuum bag, followed by the sample cooked using conventional boiling, boiling in a steam cooker, microwave heating, conventional heating, and the fresh sample in descending order.

## Figures and Tables

**Figure 1 plants-13-01283-f001:**
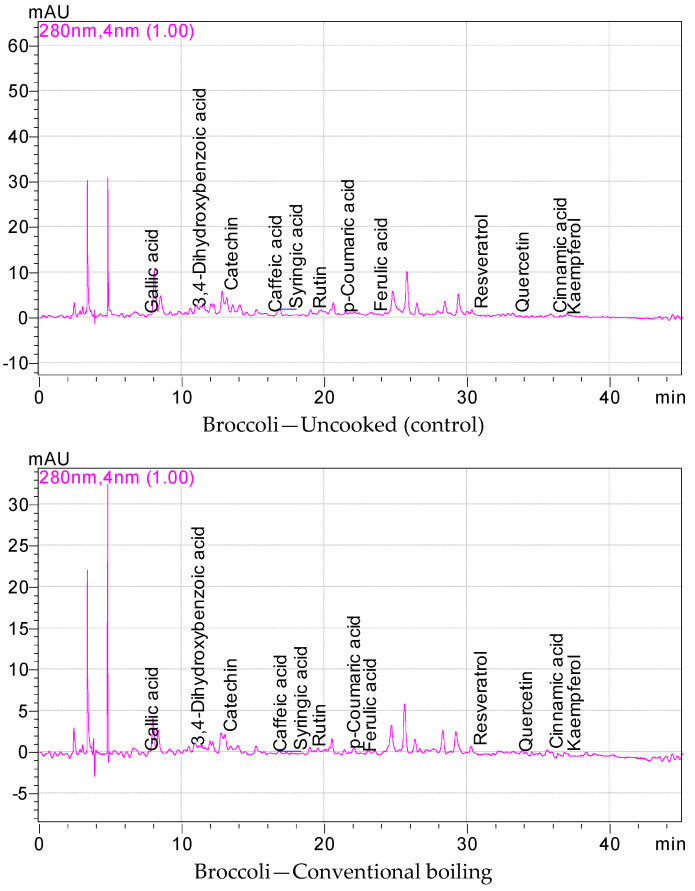
Phenolic chromatograms of broccoli samples.

**Table 1 plants-13-01283-t001:** Bioactive properties of broccoli cooked using different techniques.

Process	Moisture Content (%)	Total Phenolic Content (mg/100 g)	Total Flavonoid Content (mg/100 g)	Antioxidant Activity (mmol/kg)
Fresh	86.93 ± 0.09 d *	116.69 ± 0.74 c	157.30 ± 5.06 b	4.75 ± 0.02 c
Conventional boiling	91.82 ± 0.50 a **	36.32 ± 0.46 f	60.16 ± 0.90 f	2.55 ± 0.10 f
Boiling in a vacuum bag	89.58 ± 0.05 c	75.63 ± 1.24 d	92.86 ± 0.78 d	3.75 ± 0.05 d
High-pressure boiling	90.41 ± 0.70 b	65.05 ± 1.95 e	73.81 ± 0.78 e	3.60 ± 0.04 e
Conventional heating	72.84 ± 0.68 e	148.04 ± 4.32 b	147.78 ± 2.50 c	4.99 ± 0.00 a
Microwave heating	50.84 ± 1.81 f	423.39 ± 3.96 a	409.68 ± 3.67 a	4.79 ± 0.00 b

* Standard deviation; ** values within each column followed by different letters are significantly different: “*p* < 0.05”.

**Table 2 plants-13-01283-t002:** Phenolic compounds of broccoli cooked using different techniques.

Phenolic Compounds (mg/100 g)	Fresh	Conventional Boiling	Boiling in a Vacuum Bag	High-Pressure Boiling	Conventional Heating	Microwave Heating
Gallic acid	1.53 ± 0.43 c *	0.52 ± 0.01 f	1.91 ± 0.57 b	1.02 ± 0.36 d	0.66 ± 0.22 e	4.29 ± 0.94 a
3,4-Dihydroxybenzoic acid	3.48 ± 0.36 b **	0.41 ± 0.10 f	2.13 ± 0.26 d	1.77 ± 0.22 e	2.83 ± 0.91 c	15.17 ± 0.21 a
Catechin	24.00 ± 2.11 b	2.24 ± 0.49 f	10.93 ± 1.48 d	7.01 ± 1.43 e	13.36 ± 2.69 c	54.48 ± 0.75 a
Caffeic acid	0.83 ± 0.30 b	0.13 ± 0.02 d	0.15 ± 0.06 c	0.06 ± 0.0 e	0.86 ± 0.36 b	7.07 ± 0.17 a
Syringic acid	0.60 ± 0.06 c	0.08 ± 0.02 e	0.19 ± 0.02 d	0.16 ± 0.04 d	0.92 ± 0.49 b	6.21 ± 0.17 a
Rutin	4.16 ± 0.77 b	0.55 ± 0.06 f	1.29 ± 0.17 d	0.62 ± 0.14 e	2.56 ± 0.69 c	16.33 ± 1.58 a
*p*-Coumaric acid	0.21 ± 0.06 e	0.02 ± 0.00 f	0.08 ± 0.01 d	0.06 ± 0.01 e	0.36 ± 0.14 b	2.03 ± 0.06 a
Ferulic acid	0.21 ± 0.04 d	0.07 ± 0.03 f	0.12 ± 0.02 e	0.26 ± 0.09 c	0.59 ± 0.27 b	4.72 ± 0.21 a
Resveratrol	0.44 ± 0.03 b	0.07 ± 0.01 e	0.31 ± 0.01 c	0.13 ± 0.03 d	0.46 ± 0.08 b	1.11 ± 0.22 a
Quercetin	0.67 ± 0.04 b	0.62 ± 0.11 c	0.50 ± 0.08 d	0.40 ± 0.05 e	0.38 ± 0.10 f	4.68 ± 0.52 a
Cinnamic acid	0.16 ± 0.03 d	0.15 ± 0.03 d	0.09 ± 0.03 e	0.30 ± 0.04 c	0.33 ± 0.04 b	0.78 ± 0.04 a
Kaempferol	0.23 ± 0.06 e	0.31 ± 0.08 d	0.21 ± 0.06 ef	0.51 ± 0.06 c	0.74 ± 0.02 b	5.12 ± 0.47 a

* standard deviation; ** values within each row followed by different letters are significantly different at *p* < 0.05.

**Table 3 plants-13-01283-t003:** Mineral (mg/kg) and crude protein (%) contents of broccoli cooked using different techniques.

Treatments	P	K	Ca	Mg	S	Fe	Cu	Mn	Zn	B	Protein
Control (Fresh)	3114.94 ± 21.59 f	29,730.49 ± 299.48 c *	3555.67 ± 54.46 f	1197.02 ± 50.90 f	3935.51 ± 34.56 e	46.60 ± 1.40 e	2.73 ± 0.03 d	9.61 ± 0.55 f	11.88 ± 0.11 f	2.76 ± 0.04 d	13.35 ± 0.32 f
Conventional boiling	4383.36 ± 159.37 a	19,638.27 ± 526.42 f **	6052.69 ± 150.03 a	1610.79 ± 37.84 b	3167.03 ± 43.75 f	51.59 ± 1.14 c	3.28 ± 0.34 b	13.07 ± 0.54 e	13.48 ± 0.09 e	2.21 ± 0.41 e	15.75 ± 0.43 b
Boiling in a vacuum bag	3787.59 ± 30.86 d	32,729.06 ± 998.34 a	4731.34 ± 240.78 c	1508.55 ± 30.02 c	4208.92 ± 60.08 c	41.46 ± 0.24 f	3.55 ± 0.10 a	14.54 ± 0.05 d	16.30 ± 0.02 b	6.13 ± 0.72 a	17.27 ± 0.47 a
High-pressure boiling	3908.20 ± 16.18 c	21,876.31 ± 463.36 e	5598.51 ± 32.73 b	1326.53 ± 0.88 e	4015.41 ± 19.26 d	53.06 ± 2.11 b	2.97 ± 0.40 c	15.22 ± 0.13 a	14.34 ± 0.12 d	1.70 ± 0.02 f	15.61 ± 0.12 c
Conventional heating	3658.44 ± 85.05 e	28,322.47 ± 1190.18 d	3791.24 ± 127.93 e	5039.27 ± 557.90 a	4969.11 ± 137.71 b	55.99 ± 5.31 a	2.56 ± 0.11 f	14.89 ± 1.05 c	17.53 ± 1.16 a	5.12 ± 0.22 b	13.85 ± 0.83 e
Microwave heating	3934.30 ± 51.88 b	30,420.44 ± 211.57 b	4156.85 ± 73.95 d	1392.06 ± 42.53 d	5112.97 ± 70.60 a	48.51 ± 0.36 d	2.64 ± 0.13 e	15.04 ± 0.71 b	15.39 ± 1.71 c	4.41 ± 0.68 c	14.02 ± 0.80 d

* standard deviation; ** values within each column followed by different letters are significantly different at *p* < 0.05.

## Data Availability

Data supporting the results of this study are available from the corresponding author upon reasonable request.

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
