# Peer review of "Quantitative Determination of Biogenic Element Contents and Phytochemicals of Broccoli (Brassica oleracea var. italica) Cooked Using Different Techniques"

_plants, 2024, doi:10.3390/plants13101283_

Round 1
Reviewer 1 Report
Comments and Suggestions for Authors
Please see attachment.

Author Response
I carefully revised my article according to comments of reviewer. I pointed out corrections with red in text
Reviewer 2 Report
Comments and Suggestions for Authors
The manuscript titled "Quantitative determination of the content of biogenic and phytochemical elements of broccoli (Brassica oleracea var. italica) cooked using different techniques" in my opinion is not so novel research, since the same authors demonstrate with the references that there is a lot of research and information on broccoli and vegetable cooking techniques. The information they present is not highlighted with a good discussion of the results obtained, which is what could make it interesting for readers. In general, in the results and discussion it is necessary to improve the wording of the results obtained, denoting where differences were found without having to repeat all the numerical values of each variable analyzed. In addition, they should improve the discussions by explaining in greater depth what the differences between each cooking method in some variables can be attributed to. For example, it would be interesting if they explained what these differences between protein percentages depending on cooking technique are or are due to. It also has many spelling errors that should be checked carefully before uploading it for review for possible publication. This in a general way, but specifically I point out some observations that must be taken into account to improve the quality of the work and make it attractive:
The introduction section is very extensive and more than 70% of the references used to support it are more than 10 years old, so I suggest that you be more specific and update the references.
L44: "affects." word out of place. Correct wording.
L78-85: This paragraph is irrelevant to this manuscript since they do not address drying broccoli here. They should remove it.
L97-100: Repetitive information, you can delete the first sentence (Broccoli samples were cooked in conventional boiling, boiling in vacuum bag, highpressure boiling, conventional heating and microwave heating.).
L101: "special vacuumable plastic bag" It is important that they describe the characteristics of the bags.
L107, 124: "Camples" "Samoles" correct spelling error.
L108: "The moisture results of amaranth seed samples" amaranth seed??? explain or correct that.
L111: It is important that you describe the number of the method used.
L115: "3 g powdered broccoli samples" must explain the procedure used to turn the broccoli into powder.
L115: "20 ml" the unit of measurement for liters is L, so you must change it to 'mL' or if applicable to 'μL'. Review throughout the document.
L133-136: It must clearly describe the make and model of the equipment used, as well as all the conditions of the run (phases, flow, etc.).
L140: You must also describe all the conditions of the analysis.
L143: "among values of control and dehydration types" Explain what you mean by this.
L148: "tesults" correct spelling error.
L181-184: irrelevant information for this section, they must be changed to the respective section (3.3).
L180-200: In my opinion it is unnecessary to express here the numerical results of all the antecedents, the writing should be improved to explain only the similar or different effects with the other authors. And better explain why these differences in the results obtained by the cooking techniques of this manuscript are due.
Table 1: You must remove the asterisks (* and **) from the table and describe in the table footer that 'the results are expressed as average ± standard deviation'. and the probability value symbol must be lowercase: "p < 0.05''.
L222-229: I suggest that you eliminate the numerical values from the text and improve the writing by explaining the significant differences between the cooking techniques, since it is tiring and repetitive to read them as you have the values in a very close table.
L242-251: same comment, the numerical results reported in each reviewed manuscript should not be repeated. They only have to explain whether there is a coincidence or not with the results of this investigation.
Figure 1: what is the point of presenting figure 1 if you don't call it in the body of the document? Please take it into account in the text.
L320-323: It is not a conclusion. You should delete this sentence.
L323-324: This information should also be eliminated from this section because it is too well known since they demonstrate it with all the literature review they present.
L334-338: There are two repetitive paragraphs, eliminate one of the two.
Author Response
The points mentioned have been improved and highlighted in red
Reviewer 3 Report
Comments and Suggestions for Authors
First of all, I would like to thank the Editor for his confidence in hearing me out on the merits of the study for publication in the journal Plants. Congratulations to the authors for proposing and evaluating a large number of compounds that are important for characterizing the quality of broccoli after heat application.
It is important to point out in advance of my comments that I did not run the manuscript through a plagiarism detection program.
I made my comments directly in the text.
The work is well written and adequately justified. However, I have suggested changing the order in which the information is presented in the Introduction. I have a question about the Material and Methods section. The results are adequately presented, but they need to be better presented by dividing them into smaller paragraphs.
In the discussion, I missed the explanation of why the conventional and microwave cooking methods promote a strong decrease and increase in bioactive compounds, respectively. Despite this issue, the aim of the study was meet.

Author Response
The mentioned issues have been revised step by step and highlighted in red. Additionally, the revised article was sent as an attached file. Response letter below please:
Manuscript ID: plants-2959882
Dear Editor,
(Plants)
I carefully revised my article “Quantitative determination of biogenic element contents and phytochemicals of broccoli (Brassica oleracea var. italica) cooked by different techniques” according to comments of reviewers.
Please find my revised manuscript.
First of all, I would like to thank the Editor for his confidence in hearing me out on the merits of the study for publication in the journal Plants. Congratulations to the authors for proposing and evaluating a large number of compounds that are important for characterizing the quality of broccoli after heat application.
Answer: Thank you very much for your valuable opinions and kind words about my article.
It is important to point out in advance of my comments that I did not run the manuscript through a plagiarism detection program.
Answer: The article was checked with the iThenticate program before being sent to Plants magazine.
I made my comments directly in the text.
The work is well written and adequately justified. However, I have suggested changing the order in which the information is presented in the Introduction. I have a question about the Material and Methods section. The results are adequately presented, but they need to be better presented by dividing them into smaller paragraphs.
Answer: The issues you mentioned above in the text have been corrected step by step. Comments were made on some of them. I hope we agree with these comments. Thank you very much for your suggestions and thank you very much for your decision.
In the discussion, I missed the explanation of why the conventional and microwave cooking methods promote a strong decrease and increase in bioactive compounds, respectively. Despite this issue, the aim of the study was meet.
Answer: Thank you very much for your positive opinion.
-In abstract: One of the “between” has been removed.
- Brassicaceae was added instead of Cruciferae.
- Sentence starting with since has been revised as” Broccoli contains significant amounts of health-beneficial compounds, which has increased its consumption by people”
- The part starting with domestic was turned into a paragraph and the sentence starting with domestic was revised.
- "However, steaming has been reported to be more effective for certain health-promoting compounds" was revised "However, steaming has been reported to be more beneficial for certain health-promoting compounds in fruits and vegetables."
-“ Broccoli, which is cooked using various cooking methods can be consumed fresh or cooked in salads or garnishes, or it can be consumed independently.” Was changed as “ Broccoli cooked with various cooking methods can also be used fresh in salads or side dishes.”
-“Several heat treatments are commonly applied to inactivate microorganisms and enzymes, increase palatability, and neutralize anti-nutritive compounds in green leafy vegetables that are widely consumed raw or cooked/processed, resulting in increased product safety and quality” was changed as “Various heat treatments are widely applied to green leafy vegetables, which are commonly consumed raw or cooked/processed, to neutralize microorganisms and enzymes and increase flavor, thus increasing product safety and quality.”
-The paragraph between Line 64-70 can be the first paragraph. I am extremely respectful of your decision on this matter. However, since it does not directly describe broccoli and describes general vegetables, I think it would be more appropriate to include it here.
-I believe that it would be appropriate to include this paragraph (line72-87) here since it is considered as a hypothesis before the purpose of the study. Because the previous paragraphs gave general information about the introduction of broccoli, bioactive components and phenolics. Here, the purpose is given by giving information about the actual heat treatments applied. I hope we agree with this suggestion. I respect and thank you very much for your decision.
-What I mean by crumbled is: divided into florets. I added to text.
-Line 97-98:it was revised as “ Broccoli samples were cooked separately using conventional boiling, vacuum bag boiling, high pressure boiling, conventional heating and microwave heating methods.”
- Line 99-100: This sentence was deleted.
Line 102-105: This paragraph was revised
- After Line 148, paragraphs were created according to the flow of the subject.
-Line 148: tesults was corrected as results.
All corrections were pointed out with red color in text.
Thank you for your kind helps and contributions.
Looking forward to hearing from you.
Best wishes,
Dr Mehmet Musa Özcan
Corresponding author

Round 2
Reviewer 1 Report
Comments and Suggestions for Authors No substantial changes have been made to the work as requested.Author Response
Please see attachment below, thank you.

Reviewer 2 Report
Comments and Suggestions for Authors
The second version of the manuscript titled "Quantitative determination of the content of biogenic and phytochemical elements of broccoli (Brassica oleracea var. italica) cooked using different techniques" remains almost the same as the first version, I do not understand why they did not attend to any of the observations that I did them in the first review. For example, I suggested that they update the bibliographic references or justify why it is important for them to maintain old references. Discussions should also be improved to give greater importance to the results obtained. Nor did they address the observations in the first review that I pointed out in lines 101, 107, 124, 108, 111, 115, the units of measurement, 133-136, etc. nor do they justify why they do not consider it important not to make the suggested changes or corrections.
They changed the wording in L92: but it leaves more doubts, for example it is important that they explain in detail what "divided into florets" consists of.
For me it is still a manuscript that has no scientific relevance that could draw the attention of the readers of this Journal.
Author Response
Response for rev.2:
Answer:
1-Your comments are really important to me. Because it adds a different quality to my article. That's why I tried to revise it as much as possible as you wanted. Thank you very much for your contributions and time.
2-Reviewer 1's comments were also requested by this reviewer (Rev2). Therefore, the corrections I made for reviewer 1 are valid for reviewer 2. But Reviewer 2 has additional comments below:
The second version of the manuscript titled "Quantitative determination of the content of biogenic and phytochemical elements of broccoli (Brassica oleracea var. italica) cooked using different techniques" remains almost the same as the first version, I do not understand why they did not attend to any of the observations that I did them in the first review. For example, I suggested that they update the bibliographic references or justify why it is important for them to maintain old references.
Answer: Some of the old references have been removed. In this revision, I zoomed out as you suggested. Instead, I added new references and highlighted them in red in the reference list.
Discussions should also be improved to give greater importance to the results obtained.
Answer: The discussion section has been improved a little more according to your suggestion.
Nor did they address the observations in the first review that I pointed out in lines 101, 107, 124, 108, 111, 115, the units of measurement, 133-136, etc. nor do they justify why they do not consider it important not to make the suggested changes or corrections.
Answer: Since these units of measurement are generally the same as those in the references, no changes were made to make their interpretation more understandable.
They changed the wording in L92: but it leaves more doubts, for example it is important that they explain in detail what "divided into florets" consists of.
Answer:
For me it is still a manuscript that has no scientific relevance that could draw the attention of the readers of this Journal.
Answer: What I mean here is that the floret part was subjected to heat treatment along with the flowers, flowers and flower stems. This is already evident from the image in the graphical abstract. I hope I expressed it correctly this time.
As a result, I have highlighted the comments made in the discussion section in red.

Round 3
Reviewer 1 Report
Comments and Suggestions for Authors
Please correct 3.2.2. Determination of moisture content...
Reviewer 2 Report
Comments and Suggestions for Authors
In this version of the manuscript titled "Quantitative determination of biogenic element contents and phytochemicals of broccoli (Brassica oleracea var. italica) cooked by different techniques" the changes made improved the understanding of the manuscript, although there are some minor spelling errors that should be revised before your publication. In my opinion, most of the observations I made were attended to, for my part it can be published in this Journal.